# Robust Soliton Distribution-Based Zero-Watermarking for Semi-Structured Power Data

Lei Zhao [1], Yunfeng Zou [1], Chao Xu [1], Yulong Ma [1], Wen Shen [2,*], Qiuhong Shan [3], Shuai Jiang [3], Yue Yu [3], Yihan Cai [3], Yubo Song [3,4] and Yu Jiang [3,4]

1  State Grid Jiangsu Marketing Service Center, Nanjing 210003, China; zhaolei@sgcc115.wecom.work (L.Z.); 13814099766@163.com (Y.Z.); mmxcan@163.com (C.X.); mayulong0107@126.com (Y.M.)
2  State Grid Smart Grid Research Institute Co., Ltd., Nanjing 210003, China
3  School of Cyber Science and Engineering, Southeast University, Nanjing 210003, China; qiuh_shan@163.com (Q.S.); 220224992@seu.edu.cn (S.J.); 220224906@seu.edu.cn (Y.Y.); 220224837@seu.edu.cn (Y.C.); songyubo@seu.edu.cn (Y.S.); jiangyu@seu.edu.cn (Y.J.)
4  Purple Mountain Laboratories, Nanjing 211189, China
*  Correspondence: shenwen@geiri.sgcc.com.cn

**Abstract:** To ensure the security of online-shared power data, this paper adopts a robust soliton distribution-based zero-watermarking approach for tracing semi-structured power data. The method involves extracting partial key-value pairs to generate a feature sequence, processing the watermark into an equivalent number of blocks. Robust soliton distribution from erasure codes and redundant error correction codes is utilized to generate an intermediate sequence. Subsequently, the error-corrected watermark information is embedded into the feature sequence, creating a zero-watermark for semi-structured power data. In the tracking process, the extraction and analysis of the robust zero-watermark associated with the tracked data facilitate the effective identification and localization of data anomalies. Experimental and simulation validation demonstrates that this method, while ensuring data security, achieves a zero-watermark extraction success rate exceeding 98%. The proposed approach holds significant application value for data monitoring and anomaly tracking in power systems.

**Keywords:** robust zero-watermarking; data security; semi-structured data; text watermark; error correction coding; power data protection; robust soliton distribution

## 1. Introduction

In the field of electric power, the integrity and security of data are crucial for the stable operation of power systems. However, online collaboration channels in power marketing often support a vast number of user transactions. With the comprehensive promotion of Marketing 2.0 in the future, the integration of all online collaboration channel operations into an omnichannel business introduces increased complexity and variability in the business processes. Consequently, data security issues and threats become more prominent and severe [1]. One major concern is the need for power business systems to provide interface data for interaction with third-party terminals, posing a risk of data leakage. If sensitive data is compromised in certain critical business interfaces, the structure and functionality of the entire business activity chain can be severely impacted. Therefore, it is crucial to identify key nodes in the business chain, define protection priorities, and swiftly and accurately trace the source of data leakage in the event of anomalies such as data breaches.

To address the challenge of tracking data anomalies, researchers have actively explored various methods and technologies. These include machine learning-based data anomaly detection methods [2], such as Support Vector Machines (SVM), decision trees, random forests, or deep neural networks, which involve learning and modeling power data.

Statistical methods for anomaly detection [3,4], relying on statistical principles and models like outlier detection and probability distribution, compare the actual observed values of power data with expected values to detect anomalies. Time series analysis methods [5,6] can identify periodic and trending changes in power data. Additionally, data visualization techniques map power data into a graphical space, and anomaly detection methods [7–9] assess anomalies based on the position and distribution of data points in the graphical space. However, these methods require extensive data learning, and since power data has confidentiality requirements, they are not suitable for tracking anomalies in power data.

Furthermore, in recent years, digital watermarking technology has found widespread applications in areas such as data protection and data source authentication, and is extensively used for data tracing. Data watermarking involves embedding watermark information into data, akin to digital signatures or markers, with the aim of providing data integrity, source authentication, copyright protection, and preventing unauthorized data replication or tampering.

In the realm of digital watermarking technology, text watermarking is a topic that has received comparatively less attention in discussions about information hiding. Yet, power data often contains a substantial amount of textual data. Research on text watermarking dates back to 1997, when several text watermarking methods were proposed. Early text watermarking methods included structure-based watermarking, grammar-based watermarking, semantics-based watermarking, and image-based watermarking. It is crucial to control the intellectual property of textual content by analyzing the implementational nature and underlying logical principles of text content. The differences in text watermarking methods also impact the effectiveness of protecting digital text [10]. In structure-based watermarking, lines, letters, and spaces in the watermark carrier text are shifted to embed watermark bits [11]. In grammar-based watermarking, the grammatical structure of the text is utilized for watermark embedding. Mikhail J. Atallah and others first proposed a natural language watermarking scheme using syntax trees and transformations applied to syntax trees of structured text in 2000, preserving all properties of the text [12,13]. Semantic-based watermarking primarily uses the semantic content of the text for embedding watermarks. Atallah et al. first introduced a semantic watermarking scheme in 2000 [12,14,15]. However, the functionality of semantic-based text watermarking relies on language features, and techniques based on synonyms lack resilience against random synonym substitution attacks. In image-based text watermarking methods, text images serve as the source for watermark embedding. However, text watermarking algorithms using binary text images lack robustness against re-typing attacks and have limited applicability [16]. While text image authentication is straightforward, considering text as an image is often impractical.

Nevertheless, there has been limited work on watermark injection for semi-structured textual data, despite the fact that textual content in power data is often transmitted in a semi-structured form. Semi-structured data offers flexibility and scalability, effectively handling various data types and formats, providing convenience for data analysis and processing [17,18]. Some typical applications of semi-structured data include Extensible Markup Language (XML), JavaScript Object Notation (JSON), and others.

The aforementioned text watermarking schemes are generally not applicable or effective for semi-structured textual data. Moreover, these schemes are often not robust enough against typical attacks, such as deletion. Additionally, when protecting power data, the specific meaning of the data is crucial, and any modification or tampering would render the data invalid. Traditional text watermarking methods mentioned above typically involve modifying the data itself to embed watermark information. Therefore, there is a need to propose a robust watermarking scheme for protecting semi-structured data without modifying the data content.

In summary, this paper addresses the challenge of ineffective traceability caused by localized changes in power data. Through an analysis of power data and its circulation scenarios, we propose a semi-structured power data tracking scheme based on robust zero-watermarking. The main contributions are as follows:

(1) For the format of power data, we propose a novel scheme for robust zero-watermark embedding and extraction in semi-structured data. This scheme embeds watermark information into the feature sequence of semi-structured data without modifying the data, ensuring data integrity and privacy.

(2) The scheme integrates erasure codes and redundant error correction codes theory. The watermark to be embedded is divided into blocks, and a transfer matrix is used to obtain an intermediate sequence. Subsequently, error correction coding is applied to the intermediate sequence. Finally, the encoded watermark information is embedded into the feature sequence. The encoded watermark can detect block damage, and even after losing damaged blocks, it can still recover the original watermark information, significantly enhancing robustness.

(3) Experimental results demonstrate that our scheme ensures data integrity and exhibits high robustness, security, and accuracy.

The rest of this paper is organized as follows: Section 2 introduces the basic concepts of zero-watermarking technology. Section 3 establishes the system model of the semi-structured power data tracking scheme based on robust zero-watermarking, providing detailed explanations of power data preprocessing, and the embedding and extraction methods of robust zero-watermarking. Section 4 conducts experimental and analytical demonstrations of our proposed scheme. Finally, Section 5 provides a brief summary.

## 2. The Concept of Zero-Watermarking

Zero-watermarking technology, as a means of data labeling and authentication, has found widespread applications in the field of information security and is well-suited for protecting power data. Traditional watermarking algorithms struggle to balance the invisibility and robustness of watermarks simultaneously. Wang et al. [19] introduced the concept of "zero-watermarking". Unlike traditional watermarking algorithms, zero-watermarking algorithms construct watermarks using rich and robust feature information of the protected target data, without embedding any watermark information in the original data. Zero-watermarking technology requires the involvement of a trusted third-party entity or an entity with equivalent functionality for watermark ownership authentication and verification. When applying robust zero-watermarking technology to data anomaly tracking, in cases of data tampering or leakage, the trusted third party or an equivalent entity can leverage its credibility, invoke the authenticated and stored zero-watermark, and compare it with the anomalous data to determine the watermark information at the time of the data anomaly, swiftly and accurately tracing the source of the data anomaly [20].

In zero-watermarking schemes, the watermark information is not actually embedded in the original data itself; instead, the watermark carrier uses certain features of the original data for generation. These features remain stable before and after the transmission of the original data to ensure the effectiveness of the zero-watermark extraction process.

Zero-watermarking schemes comprise two main phases: zero-watermark embedding and zero-watermark extraction and verification. Watermark embedding is performed by the watermark owner and is then released on a trusted third-party certification entity. During watermark extraction, the verifier requests the stored zero-watermark from the trusted third-party certification entity and combines the features of the data in their possession to complete the watermark extraction process, verifying the watermark owner and the data source. In this algorithm, a trusted certification entity is a fundamental requirement for the original data owner to register their legitimate identity. When there are issues with data ownership or watermark source, this trusted third party serves as a decision authorization entity.

The general process of zero-watermark embedding is illustrated in Figure 1. Firstly, the data owner extracts the robust feature information of the data to be protected and obtains feature values based on these features. These feature values are then encoded into a feature sequence capable of embedding watermark information. Next, the data owner selects specific watermark information as proof of identity and encodes this infor-

mation. The encoded feature sequence is used as the watermark carrier, and the encoded watermark is embedded, resulting in the zero-watermark sequence for the data. Finally, the zero-watermark sequence and the data to be protected are stored on a trusted third-party certification entity or an entity with equivalent functionality for subsequent identity verification of the watermark owner.

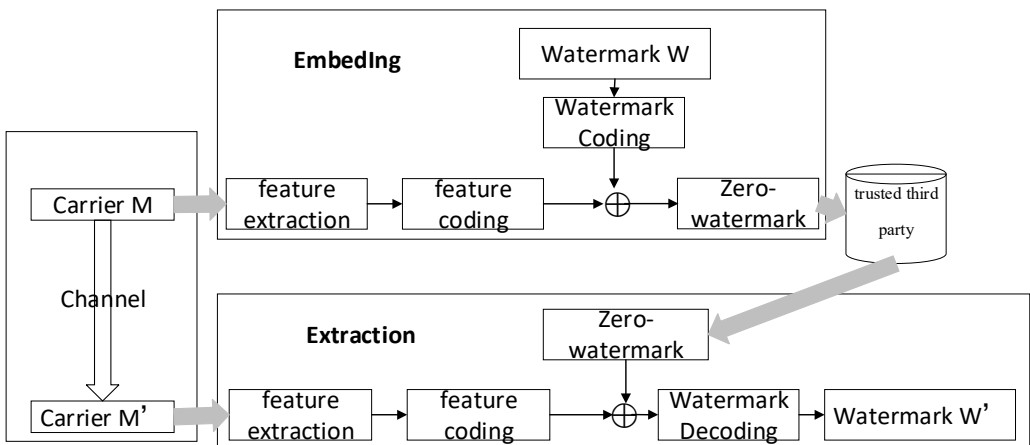

**Figure 1.** The schematic diagram of the zero-watermark embedding and extraction process.

The main idea behind zero-watermark extraction is to combine the zero-watermark sequence stored in the trusted third-party certification entity or an equivalent entity with the feature sequence extracted from existing data through a series of operations to recover the watermark information. This process verifies the identity of the data owner. The general process of zero-watermark extraction and verification is depicted in Figure 1. Firstly, the robust feature information of the data to be verified is extracted, and feature sequences are obtained based on these features. The feature sequence is then combined with the zero-watermark sequence stored in the trusted third-party certification entity or an entity with equivalent functionality through the inverse operation of embedding. This process recovers the watermark information. Finally, the original watermark information stored in the trusted third-party certification entity or an entity with equivalent functionality is compared with the recovered watermark information for similarity. This determines the identity of the data owner, completing the extraction and verification of the watermark.

### 3. Robust Soliton Distribution-Based Zero-Watermark Scheme for Semi-Structured Power Data

The proposed semi-structured power data tracking scheme based on robust zero-watermarking extracts the watermark sequence from the corresponding zero-watermark stored in a trusted third-party entity when data leakage occurs. This process enables the identification of nodes through which the data has passed, ultimately locating the source of data leakage.

Before embedding the watermark information into the watermark carrier, preprocessing is performed on both. Feature extraction is applied to the original semi-structured power data to construct a feature sequence, serving as the watermark carrier and identifying effective embedding positions. The watermark information is then segmented, and a transmission matrix is constructed to transform these segmented fragments into an intermediate sequence. These intermediate sequences undergo further encoding through error correction coding to resist a variety of attacks, enhancing robustness. The encoded sequences are then embedded into the watermark carrier, generating the zero-watermark sequence. The zero-watermark sequence is stored in a trusted third-party entity. During the zero-watermark extraction and verification at the receiver end, successful extraction is possible even if the received zero-watermark sequence has been intentionally or unintentionally modified.

In the context of power business data, the majority of transmitted data adopts the format of semi-structured data. Semi-structured data includes formats such as XML, JSON, and CSV. JSON is the most common among them. For simplicity, this paper represents the implementation and experimental verification of the proposed scheme using JSON format. Experimental results demonstrate the scheme's robustness against various typical attacks. Additionally, the proposed scheme can be easily extended to other typical semi-structured data formats, such as tabular data formats like Microsoft Excel or comma-separated values (CSV).

### 3.1. Data Anomaly Tracking Scheme

The power data involves some privacy information of power customers and labels added by the power company to identify specific transactions. As such, it falls under non-public data. After its formation, this data needs to circulate among various departments within the power company. Therefore, in the event of data tampering or leakage, it is crucial to track the anomalous data. This paper proposes a data anomaly tracking scheme utilizing zero-watermarking technology to achieve the identification of the data owner and facilitate tracking.

Figure 2 illustrates the schematic diagram of the proposed scheme. Before sending the data to the next circulating node, the original data owner generates a zero-watermark sequence containing identity information and associates it with their original data. This zero-watermark sequence is stored in a trusted third-party authentication entity or an equivalent entity. Since the watermark is independent of the data file itself, it does not affect the content of the file. Upon receiving the data, the next circulating node can request the zero-watermark sequence from the trusted third-party authentication entity or an equivalent entity and, in conjunction with the data, extract the watermark information to verify the data's source. After ensuring the legitimacy of the data source, the circulating node generates a zero-watermark sequence containing identity information for itself and the upper-level node and stores it in the trusted third-party authentication entity. Subsequently, the file continues to circulate to lower-level nodes.

During the data circulation process, it is possible to determine the nodes through which the data has passed by examining the zero-watermark sequences stored in a trusted third-party entity. In the event of data tampering or leakage onto the internet, all data owners can trace the origin of the current data by combining the leaked file with the zero-watermark sequence stored in the trusted third-party entity. This facilitates data anomaly tracking.

### 3.2. Preprocessing of Semi-Structured Data

Figure 3a presents an illustrative example of a JSON file. The JSON file adopts a hierarchical or parallel format for data storage. As depicted in Figure 3b, this paper represents the data hierarchy within the JSON using a tree structure. The received raw JSON-formatted power data is considered the root JSON object, and all data is stored in its leaf nodes. Elements in the JSON file can be classified into three types: JSON objects that store data hierarchically, JSON arrays that store elements in parallel, and JSON primitives directly storing data. Both JSON objects and JSON primitives consist of key-value pairs, while JSON arrays carry parallel data without having their own keys.

During the data preprocessing phase, this paper initially parses the raw power data JSON file (denoted as) as the provider of feature sequences. It is parsed into key-value pairs, which serve as the basic units for embedding watermark information in the subsequent scheme. The scheme initializes as an empty set and establishes a tree structure to represent hierarchical relationships. Traversing all nodes in the tree starting from, if a JSON object is encountered during the traversal, its key is appended to the end of the prefix, where the prefix initially is an empty string. Subsequently, an underscore "_" is appended at the end to signify the hierarchical relationship, and the traversal continues to the child nodes of

this object. In case a JSON array is encountered during parsing, its elements are accessed in parallel. If a JSON primitive is encountered, its key is appended to the prefix.

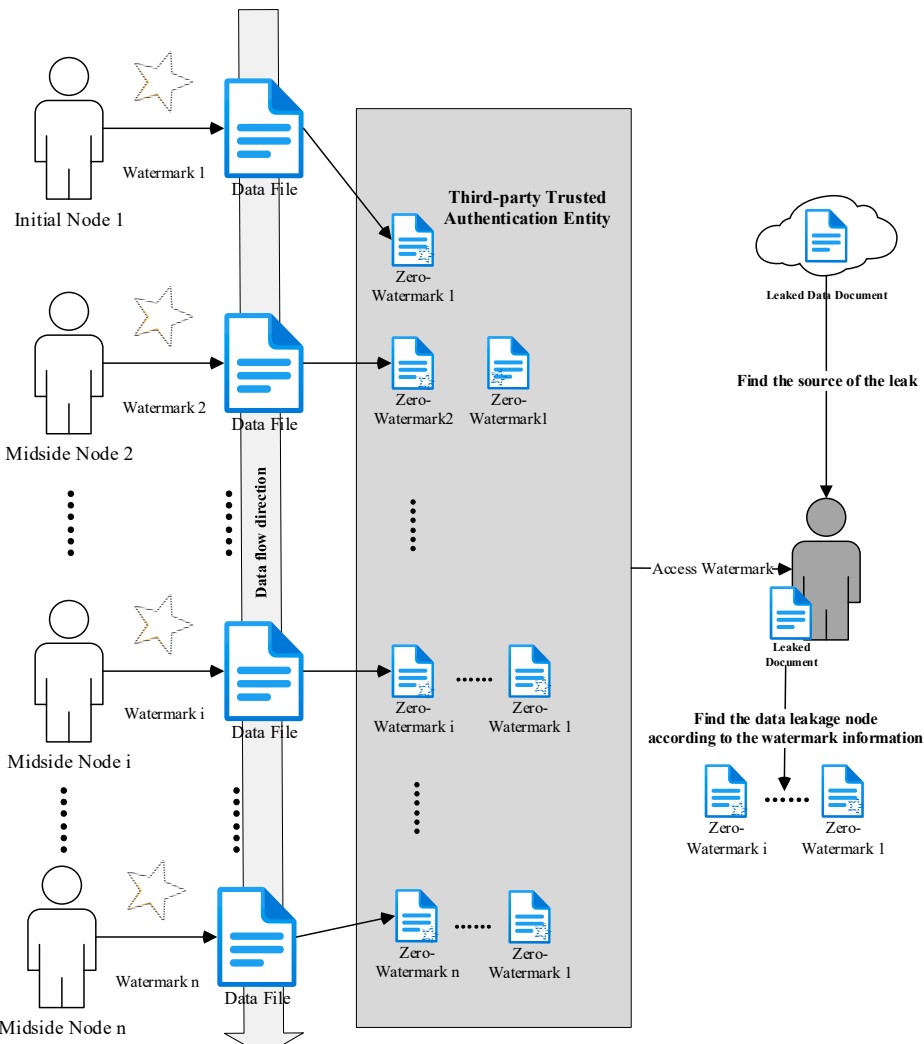

**Figure 2.** Data Anomaly Tracking.

The corresponding pairs for leaf nodes are represented as $p = \{prefix, value\}$, where the value is the value of the respective leaf node. Subsequently, the scheme performs backtracking and continues the traversal until all nodes have been visited. After parsing, the set of key–value pairs $P$ is constructed, comprising all pairs formed by the leaf nodes of the tree. Let $q$ denote the number of leaf nodes. Figure 3c illustrates the parsing result of the file depicted in Figure 3b.

In practical applications, power systems monitor and filter transmitted files using keyword filtering. However, for some plain text data, there may not be a comprehensive standard, leading to the possibility of synonymous substitutions within the text. In such cases, although the text may undergo synonymous substitutions, it essentially represents the same data. Consequently, the extracted feature sequences may exhibit expression differences. To address the aforementioned issues and considering that there might be variations in the representation of plain text data, particularly due to synonymous substitutions, this paper's approach exclusively embeds watermark data into feature sequences extracted from key-value pairs $P_{num}$, where the values consist solely of numerical data. This strategy is employed to mitigate potential semantic alterations resulting from changes in string-type data after the embedding of watermark information bits.

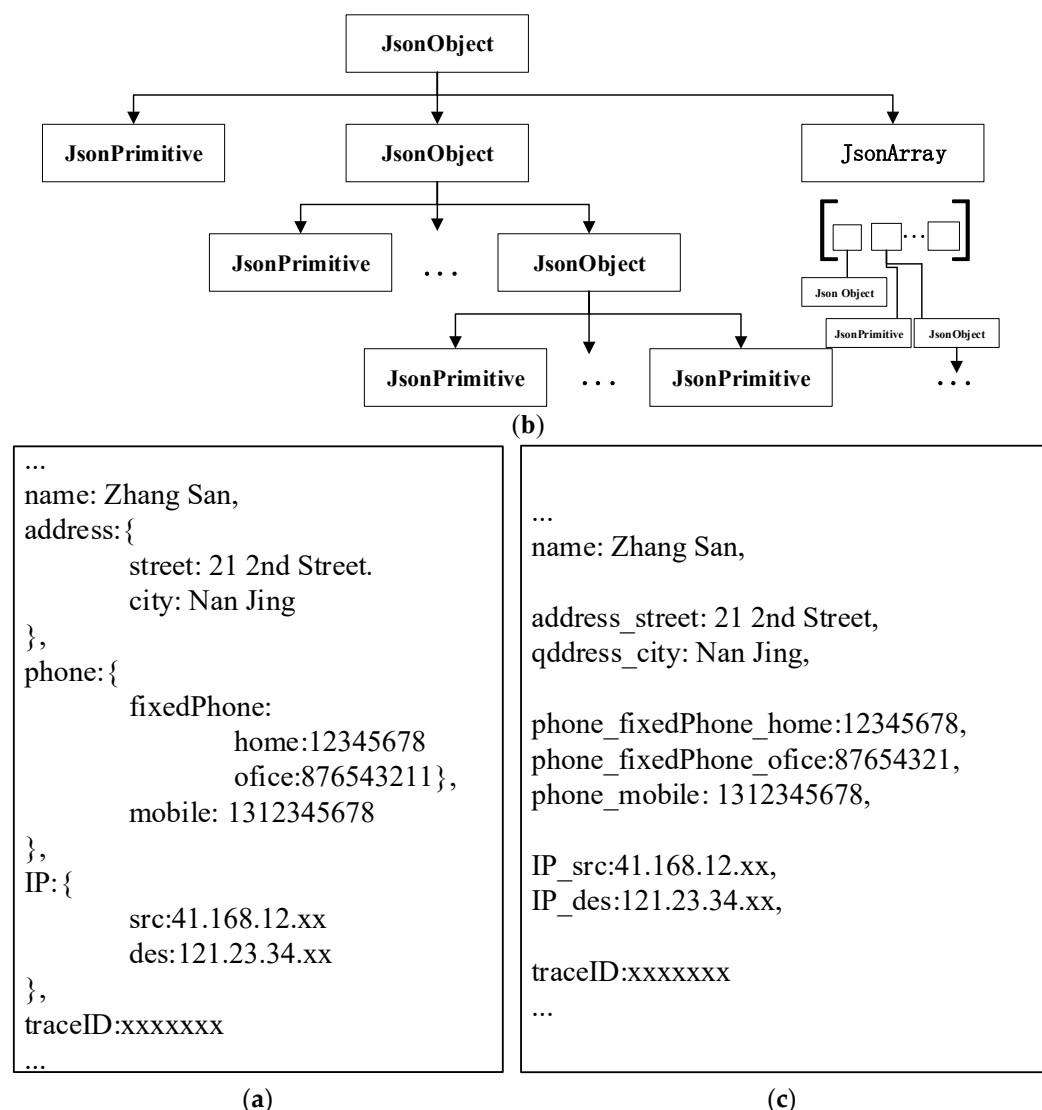

**Figure 3.** (**a**) JSON File Structure, (**b**) Raw JSON File, and (**c**) Parsed JSON Key-Value Pairs.

Furthermore, it is necessary to define a set of key–value pairs $P_{ban}$ to be excluded from those eligible for watermark embedding. For instance, information such as phone numbers, IDs, or timestamps should be excluded, as they uniquely identify the data, making it easy to reverse-engineer the original content from the constructed feature sequence. This poses a risk of privacy data leakage. Hence, we exclusively utilize the remaining pairs $P' = P_{num} - P_{ban}$ to construct the feature sequence for embedding the watermark information. We denote the valid pairs as $p'_i P', i = \{1, 2, \ldots, n\}$, where $n$ represents the size of $P'$.

Subsequently, we generate a set of sequences $S_{rand} = \{s_1, s_2, \ldots, s_n\}$, where each $s_i$ is pseudo-randomly generated with a key as the seed. The elements in $S_i$ fall within the range [0,1]. $S_i$ determines the starting position for embedding watermark bits, and the key is shared among all parties involved in the power data flow while being kept confidential. This ensures that even if the zero-watermark is leaked, existing watermark information cannot be extracted, preventing attackers from adding a newly forged watermark.

Finally, the key–value pairs in $P'$ are sorted in ascending order based on dictionary order, and the values of all valid key-value pairs are concatenated to form the watermark carrier feature sequence $Q$.

### 3.3. Robust Zero-Watermark Embedding

The watermark embedding process is illustrated in Figure 4.

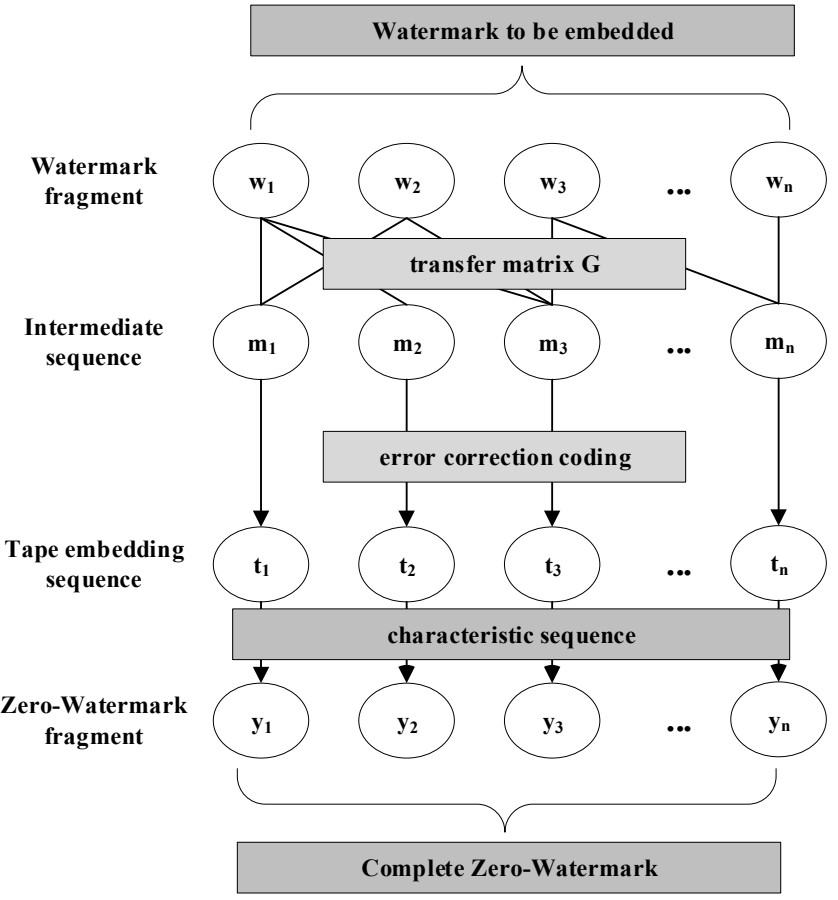

**Figure 4.** Watermark Embedding.

In the robust zero-watermark scheme proposed in this paper, the first step involves segmenting the watermark information and constructing an intermediate sequence $M = \{m_1, m_2, m_3, \ldots, m_n\}$ consisting of n segments. The length of each segment $m_i$ is p. Next, a transition matrix G is defined as follows:

$$G = \begin{pmatrix} g_{1,1} & \cdots & g_{1,n} \\ \vdots & \ddots & \vdots \\ g_{k,1} & \cdots & g_{k,n} \end{pmatrix} \qquad (1)$$

In the above, G is a sparse matrix, where each $g_{i,j}$ takes values of 0 or 1. The construction of G is performed as follows: initially, the cumulative distribution function cdfRSD [21] of the robust soliton distribution is generated. Error Correction Codes (ECC) and Robust Soliton Distribution (RSD) are two critical concepts in the realm of digital communication and data encoding. ECC encompasses a set of techniques used to detect and rectify errors in transmitted data, ensuring its integrity and accuracy over noisy channels or unreliable storage. These codes, such as Reed–Solomon and Hamming codes, add redundant data to the original message, allowing for the correction of errors at the receiver's end. While ECC focuses on error correction within a fixed-size data block, RSD optimizes the transmission process in dynamic communication environments, making them both indispensable for reliable and efficient digital communications.

The probability density $P_i$ of RSD is defined as follows:

$$P_i = \frac{\rho_i + \tau_i}{\sum_i (\rho_i + \tau_i)} \tag{2}$$

Here, ρi represents the ideal soliton distribution, and τi is defined as:

$$\tau_i = \begin{cases} \frac{R}{ik}, 1 \leq i \leq [k/(R-1)] \\ \frac{Rln(\frac{R}{\delta})}{k}, i = [k/R] \\ 0, others \end{cases} \tag{3}$$

where $R = c\sqrt{k}ln\left(\frac{k}{\delta}\right)$ represents the expected fluctuation size.

For the i-th column of G, this scheme converts a certain number of distinct $g_{j,i}$ indices from 0 to 1 based on the following algorithm:

$$\begin{cases} x_i = cdf_{RSD}(S_i[1]) \\ g_{j,i} = \begin{cases} 0, Uni(S_i[k])modn = 1 \\ 1, otherwise \end{cases} \end{cases} \tag{4}$$

Furthermore, we define the process of transferring W to M as:

$$M = W \bigoplus G \tag{5}$$

Here, $\otimes$ represents the XOR operation between matrices.

$$m_i = \sum_{j=1}^{k} w_j g_{i,j} \tag{6}$$

Then, the intermediate sequence M is generated. After processing, the original watermark information is transformed from k blocks to n blocks. During extraction, the original watermark information can be reconstructed from a subset of the new *n* blocks. For each intermediate data segment $m_i$, this paper also applies cyclic encoding as error correction to enhance the robustness of the generated watermark. The watermark information for embedding after cyclic encoding is denoted as $t_i = Cyc(m_i)$. After cyclic encoding, the length of $t$ is denoted as $s(s > p)$.

Furthermore, this paper iteratively embeds $t_i$ into the feature sequence $Q$ extracted from the original semi-structured data. Split $Q$ into n segments, forming the set $Q = \{x_1, x_2, \ldots, x_n\}$. The length q of each segment $x_i$ should satisfy q>s. Subsequently, the processed watermark information $t_i$ is embedded into each watermark carrier segment $x_i$. In this paper, the data format of $Q$ is converted from numerical to string, and the least significant bit (LSB) of each character is XORed with each bit of $t_i$ for data embedding. After embedding the watermark information $t_i$ into $x_i$, it is represented as $y_i$.

To prevent zero-watermark leakage and thwart attackers attempting to extract and construct new forged watermarks, this paper, during data preprocessing, generates a sequence $S_{rand} = \{s_1, s_2, \ldots, s_n\}$ using a confidential key. $s_i$ represents the initial position offset in bits when performing embedding. For instance, if the segment of the watermark carrier is denoted as $X$ and the sequence to be embedded is $a$, according to $s_i$, $a[j]$ will be XORed with the $(s_i + j)mod\ len\ (X)$-th bit of $X$.

$$a[j] = mod(X[s_i + j], 2), j = 1, 2, \ldots, s \tag{7}$$

Finally, the robust zero-watermark embedding process yields n zero-watermark sequences denoted as $Y = \{y_1, y_2, \ldots, y_n\}$. Concatenating these n short sequences from Y results in the final robust zero-watermark. The data owner, after generating the robust zero-watermark by combining the original semi-structured data with watermark information

identifying their own identity, uploads it to a trusted third-party certification authority or an entity with equivalent functionality.

### 3.4. Robust Zero-Watermark Extracting

When the data recipient obtains the semi-structured power data J*, slight changes may occur during the transmission process. However, due to the robustness of the data feature sequence extraction, the recipient can still request the corresponding zero-watermark Y* from a trusted third-party certification authority or an entity with equivalent functionality. This allows them to attempt to extract watermark information from the zero-watermark.

The flowchart of the extraction process is depicted in Figure 5. It is crucial to note that the data recipient needs to be aware of the key used for generating the random sequence in the zero-watermark embedding process; otherwise, watermark extraction cannot be performed.

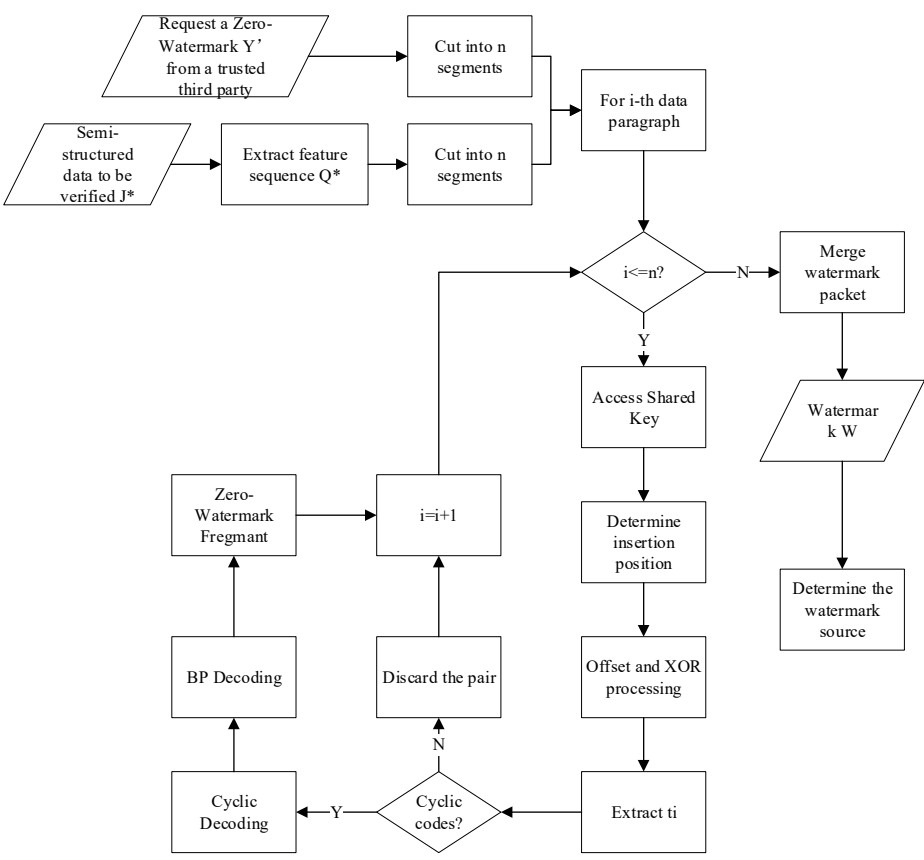

**Figure 5.** Robust Zero-Watermark Extraction.

The data recipient first performs preprocessing by generating a pseudo-random sequence $S_{rand} = \{s_1, s_2, \ldots, s_n\}$ using the key. Then, the recipient initializes G as an empty $n \times k$ matrix and begins the watermark extraction process. Then, Si[1] is utilized to obtain $x_i$, and $g_{j,i}$ is flipped using $[S_i[1] + 1, \ldots, S_i[1] + x_i]]$, as described in Section 3.2. Subsequently, they obtain the precise transmission matrix G for data embedding.

Next, the valid key-value pairs in P* are sorted in ascending order based on dictionary order. The values of all valid key–value pairs are then concatenated to form the watermark carrier feature sequence $Q*$, which is divided into n segments, constituting the set $Q* = \{x'_1, x'_2, \ldots, x'_n\}$. Simultaneously, Y* is also split into n segments, forming the set $Y* = \{y'_1, y'_2, \ldots, y'_n\}$.

If the segment of the zero-watermark sequence to be extracted is represented as $Y$, and the segment to be extracted is denoted as $a$. According to $s_i$ and $a[j]$ will be embedded into

the $(s_i + j)\,mod\,len\,(Y)$ bit of $Y$. Therefore, the recipient obtains the watermark sequence using this formula:

$$a[j] = mod(Y[s_i + j], 2), j = 1, 2, \ldots, n \tag{8}$$

Utilizing the sequence $t'_i$ with G, we need to perform cyclic decoding on $t'_i$ to recover the intermediate sequence $m'_i$. If $m'_i$ is not a valid cyclic code, the recipient considers the watermark information contained in this segment of the zero-watermark sequence to be tampered with. Therefore, the recipient will refrain from using this data packet to extract the complete watermark.

After discarding data packets with invalid cyclic codes, for the remaining valid $n'$ data packets $\{m'_1, m'_2, \ldots, m'_{n'}\}(n' \leq n)$, the data recipient further employs the Belief Propagation (BP) algorithm [22–24] for decoding. Firstly, identify the intermediate data packets that are linked to only one segment of watermark information (i.e., degree of connectivity). After recovering these intermediate data packets, leverage them to reduce the connectivity of related data packets, subsequently enabling the recovery of intermediate data packets with low connectivity. Through iteration, all intermediate data packets can be decoded. Finally, extract the complete watermark information by connecting all recovered data packets.

## 4. Experimental Results and Analysis

To validate the robustness of the zero-watermark scheme, multiple experiments were conducted on a substantial number of JSON files in this study. The experimental dataset was provided by the Southern Division of the Customer Service Center of the State Grid Corporation of China. Binary random sequences were employed as digital watermarks during the testing process. The zero-watermark in this study is composed of feature sequences extracted from JSON data, ensuring that embedding the watermark does not modify the original data. The robustness of the zero-watermark against various attacks was tested, including typical attacks such as context truncation, modification, and redundant insertion.

The generation and extraction of zero-watermarks in this study were performed on a personal laptop with a 3.20 GHz CPU and 8.00 GB of RAM.

This study tested the robustness of semi-structured data watermarking against three types of attacks, applying varying degrees of typical attacks such as deletion, value erasure, and redundant insertion. The success rate of extracting the complete watermark from zero-watermarking was assessed, and a comparison was made with scenarios where watermark information was encoded without the use of erasure codes. For a fixed watermark size of 1000 bits, this paper conducted zero-watermark embedding and extraction tests using four different-sized JSON files: File 1, File 2, File 3, File 4, and File 5, each containing 100, 200, 800, 1500, and 3000 key-value pairs, respectively, after preprocessing and parsing. Multiple identical attacks were executed on each zero-watermark file. The success rates of watermark extraction under different attacks are presented in Table 1, where extractions with incorrectly retrieved bits are not considered successful. In the table, $P_d$, $P_i$ and $P_m$, respectively, indicate the percentage of key-value pairs that were deleted, inserted, or manipulated in the file. The results demonstrate that the method proposed in this paper significantly enhances the robustness of the watermark, especially against numerical tampering attacks.

In addition, in the aforementioned File 3, File 4, and File 5, watermark information was embedded with bit sizes of 500, 1000, 1500, 2000, 2500, and 3000. The success rates of zero-watermark extraction were tested under three types of attacks: deletion, value tampering, and redundant insertion. In the tests, $P_d$, $P_i$ and $P_m$ were all set to 10%. The experimental results are illustrated in Figure 6.

**Table 1.** Extraction Success Rates of Semi-Structured Data under Different Attacks.

| Type of Attack | | Direct Extraction Success Rate/Extraction Success Rate Using Our Method | | | | |
|---|---|---|---|---|---|---|
| | | Document 1 | Document 2 | Document 3 | Document 4 | Document 5 |
| **No Attack** | | 1.0/1.0 | 1.0/1.0 | 1.0/1.0 | 1.0/1.0 | 1.0/1.0 |
| **Deleted Attack** | Pd = 5% | 0.817/1.0 | 0.820/0.975 | 0.813/0.999 | 0.821/0.998 | 0.840/0.991 |
| | Pd = 10% | 0.610/0.957 | 0.554/0.995 | 0.653/0.997 | 0.541/0.950 | 0.687/0.981 |
| | Pd = 20% | 0.361/0.967 | 0.343/0.966 | 0.299/0.996 | 0.381/0.976 | 0.374/0.972 |
| | Pd = 30% | 0.121/0.958 | 0.201/0.984 | 0.176/0.976 | 0.141/0.992 | 0.187/0.981 |
| **Modified Attack** | Pm = 5% | 0.671/0.986 | 0.487/1.0 | 0.578/0.984 | 0.398/0.990 | 0.633/0.989 |
| | Pm = 10% | 0.321/0.997 | 0.241/0.987 | 0.322/0.988 | 0.178/0.991 | 0.452/0.988 |
| | Pm = 20% | 0.167/0.998 | 0.101/0.984 | 0.221/0.979 | 0.089/0.986 | 0.281/0.995 |
| | Pm = 30% | 0.076/0.996 | 0.009/0.990 | 0.067/0.995 | 0.004/0.996 | 0.079/0.976 |
| **Inserted Attack** | Pi = 5% | 0.876/0.999 | 0.859/1.0 | 0.778/0.998 | 0.862/0.996 | 0.856/0.998 |
| | Pi = 10% | 0.675/0.988 | 0.685/0.987 | 0.574/0.997 | 0.681/0.979 | 0.591/0.997 |
| | Pi = 20% | 0.471/0.987 | 0.461/0.980 | 0.310/0.986 | 0.444/0.990 | 0.297/0.979 |
| | Pi = 30% | 0.223/0.988 | 0.291/0.987 | 0.177/0.979 | 0.187/0.984 | 0.141/0.990 |

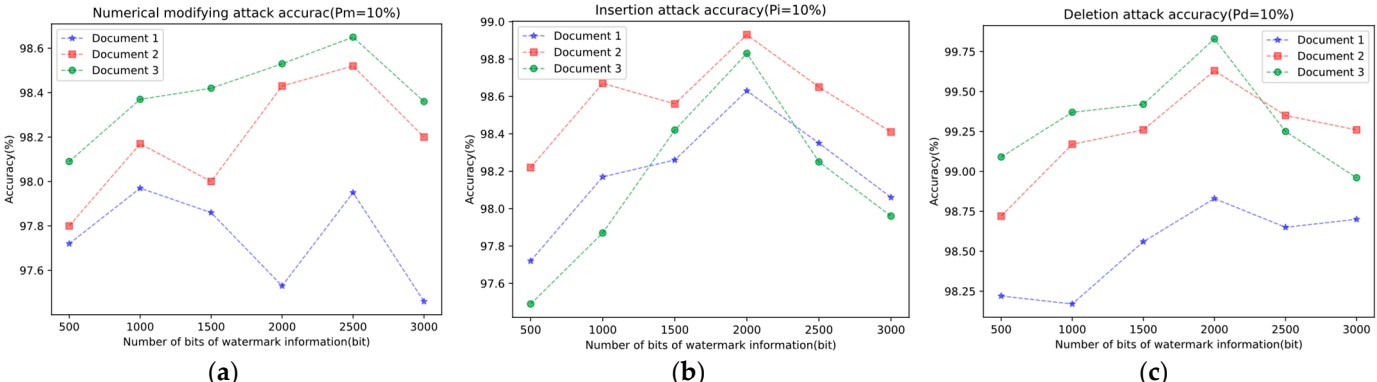

**Figure 6.** (**a**) Extraction accuracy of value modifying attacks, (**b**) extraction accuracy of insertion attacks, and (**c**) extraction accuracy of deletion attacks.

From the experimental results, it can be observed that the proposed zero-watermark generation and extraction scheme exhibit a certain level of robustness against all the mentioned attacks, even as the intensity of the attacks gradually increases. When the number of bits is too low, the robustness of the watermark after encoding is diminished. Additionally, files with more valid key-value pairs progressively demonstrate stronger robustness. The high robustness of this scheme is primarily attributed to the following reasons according to the algorithm analysis in this paper:

(1) For deletion attacks, the scheme can use the remaining valid key–value pairs to extract the remaining watermark information fragments, enabling complete watermark information recovery.

(2) For value tampering attacks, the scheme employs cyclic coding checks to identify the tampered positions and discards the tampered key–value pairs.

(3) For insertion attacks, it is challenging to extract meaningful data from the inserted pairs through cyclic coding checks; thus, they are also discarded.

In addition, from our experimental findings, it seems that a watermark bit size of around 2000 is most suitable for the method discussed in our paper. However, it is important to consider the capacity of the watermark carrier in practical applications, as it determines the size of the watermark information it can hold.

The proposed scheme also demonstrates robustness against combined attacks, as indicated in Table 2. All experimental results suggest that the scheme exhibits a certain level of reliability and is suitable for practical applications.

**Table 2.** Rate of successful extraction under different combination attacks.

| Type of Attack | Direct Extraction Success Rate/Extraction Success Rate Using Our Method | | |
| --- | --- | --- | --- |
| | Document 1 | Document 2 | Document 3 |
| (Pd = 10%, Pm = 10%) | 0.234/0.999 | 0.351/1.0 | 0.283/0.998 |
| (Pd = 10%, Pi = 10%) | 0.132/0.988 | 0.187/0.987 | 0.236/0.997 |
| (Pm = 10%, Pi = 10%) | 0.228/0.987 | 0.190/0.980 | 0.310/0.986 |

Furthermore, to enhance the security of this scheme, a key shared between the data owner and the data recipient is employed. Different keys lead to distinct starting embedding positions, aiding in preventing zero-watermark leakage and resisting collusion attacks. In other words, two data recipients cannot infer the embedding positions by comparing two identical files with different watermarks.

## 5. Conclusions

This study addresses data security concerns in the power industry and proposes a robust solitary wave distribution-based zero-watermark for semi-structured power data, enabling effective traceability in the event of localized changes in power data. Initially, feature sequences are extracted from semi-structured power data. Subsequently, these feature sequences serve as zero-watermark carriers, undergoing processing and segmentation into an equal number of blocks with the watermark body. Utilizing the robust solitary wave distribution in erasure codes and the theory of redundant error correction codes, an intermediate sequence is obtained through a transfer matrix and encoded. Finally, the error-correcting encoded watermark information is embedded into the feature sequences, generating a robust zero-watermark. During the data tracing process, effective identification and localization of abnormal data changes can be achieved by extracting and analyzing the robust zero-watermark of the traced data.

Through experimental and simulation verification, the proposed method not only ensures data security but also achieves a zero-watermark extraction success rate of over 98%. The outcomes of this study have significant practical applications in data monitoring and anomaly tracing in power systems.

**Author Contributions:** Conceptualization, L.Z.; methodology, L.Z.; writing—review and editing, L.Z. and Y.Z.; visualization, Q.S. and S.J.; supervision, Y.Y., Y.C., C.X. and W.S.; project administration, Y.M., Y.S. and Y.J. All authors have read and agreed to the published version of the manuscript.

**Funding:** This work was supported by the Science and Technology Project of State Grid Jiangsu Electric Power Co., Ltd. [5400-202318221A-1-1-ZN].

**Data Availability Statement:** We confirm that the data supporting the findings of this study are available within the article. Additional data that support the findings of this study are available from the corresponding author upon reasonable request.

**Acknowledgments:** We would like to express our gratitude to the Science and Technology Project of State Grid Jiangsu Electric Power Co., Ltd. [5400-202318221A-1-1-ZN] for their support. We would also like to acknowledge the collective efforts of all the authors in this study.

**Conflicts of Interest:** Author Wen Shen was employed by the company State Grid Smart Grid Research Institute Co., Ltd. The remaining authors declare that the research was conducted in the absence of any commercial or financial relationships that could be construed as a potential conflict of interest.

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
