# Peer review of "Robust Soliton Distribution-Based Zero-Watermarking for Semi-Structured Power Data"

_electronics, doi:10.3390/electronics13030655_

Round 1

Reviewer 1 Report

Comments and Suggestions for Authors

In this paper, a zero-watermarking method based on the robust soliton distribution is proposed, specifically used for tracing semi-structured power data. In the experiment section, the authors studied three types of common attacks faced in this application and demonstrated the robustness of the proposed method. Overall, this paper is clearly written and provides contribution. However, the reviewer has the following comments to help improve this paper.

1.        Adding references at a few places is recommended: e.g., the “Marketing 2.0” in the first paragraph as people might not be familiar with, and “re-typing attacks” also in the introduction section.

2.        Is there a formal definition of “semi-structure textual data” in the research community? If so, it should also be provided, since this is the main subject of this paper. The XML and JSON examples are pretty helpful here.

3.        Missing letter “(t)akes” below Eqn. (1).

4.        By convention, it is more widely accepted that XOR operation uses a symbol of “+” inside a circle than using “x” inside a circle. Also, do you mean “elementwise” XOR operation between matrices?

5.        It looks like in Figure 6, The extraction accuracy under all three attacks presents an upside-down U shape, could the authors explain the reason behind this? Can this be generalized and provide some guidance on how to choose the number of bits in the future applications?

6.        Could the authors provide more numerical results for the final paragraph in Section 3?

Author Response

Dear Editor-in-Chief:

Thank you for allowing a resubmission of our manuscript, with an opportunity to address the reviewers’ comments. Those comments are all valuable and very helpful for revising and improving our paper, as well as the important guiding significance to our research. Each suggested revision and comment, brought forward by the reviewers was accurately incorporated and considered. We have revised the content of the manuscript according to the valuable suggestions.

We highlighted all the revisions in red color. And the responses are highlighted in blue color.

Best regards,

Sincerely yours,

<Lei Zhao, Yunfeng Zou and Chao Xu> et al.

Response to reviewers

We gratefully thank the EIC and all reviewers for their time spent making their constructive remarks and useful suggestions, which has significantly raised the quality of the manuscript and has enabled us to improve the manuscript. Each suggested revision and comment, brought forward by the reviewers, was accurately incorporated and considered. Below the comments of the reviewers are response point by point and the revisions are indicated.

Reviewer #1:

  1. Comment: Adding references at a few places is recommended: e.g., the “Marketing 2.0” in the first paragraph as people might not be familiar with, and “re-typing attacks” also in the introduction section.

Response:

Thank you for your comment.

We have carefully revisited the introduction section of our paper and have now provided relevant references to explain terms that might be unclear to some readers. We appreciate your guidance on this aspect. Additionally, in response to the specific points mentioned in your feedback, we have also included appropriate references. In response to the concept of " Marketing 2.0", we have considered your input and concluded that it may not hold universal significance. Therefore, we have decided to remove the sentence pertaining to this concept. Your input has been invaluable in enhancing the clarity and comprehensiveness of our paper. We are committed to continuous improvement and thank you for helping us refine our work.

Author action:

Page 1: In Chapter 0, we have delete the conception of “Marketing 2.0”.

Page 2: In Chapter 0, we have provided relevant references to explain terms that might be unclear to some readers.

  1. Comment: Is there a formal definition of “semi-structure textual data” in the research community? If so, it should also be provided, since this is the main subject of this paper. The XML and JSON examples are pretty helpful here.

Response:

Thank you for your comment.

"Semi-structured textual data" refers to a type of data that does not conform to a rigid database structure but still contains tags or other markers to separate semantic elements and enforce hierarchies of records and fields within the data. This definition, however, might vary slightly among different disciplines or research communities. In the context of data management and computer science, semi-structured data is understood as a form of data that lies between structured and unstructured data. Additionally, we have included a new reference in our manuscript to further elucidate the concept in question.

Author action:

Page 2 : we have included a new reference in our manuscript to further elucidate the concept of “Semi-structured textual data”.

  1. Comment: Missing letter “(t)akes” below Eqn. (1)

Response:

Thank you for your comment.

We acknowledge the oversight and have made the necessary corrections. Thank you for bringing this to our attention, and we appreciate your understanding. This modification ensures greater accuracy and clarity in our work.

Author action:

We have made the necessary corrections.

  1. Comment: By convention, it is more widely accepted that XOR operation uses a symbol of “+” inside a circle than using “x” inside a circle. Also, do you mean “elementwise” XOR operation between matrices?

Response:

Thank you for your comment.

We acknowledge the oversight and have made the necessary corrections. Thank you for bringing this to our attention, and we appreciate your understanding. This modification ensures greater accuracy and clarity in our work.

Author action:

We have made the necessary corrections.

  1. Comment: It looks like in Figure 6, The extraction accuracy under all three attacks presents an upside-down U shape, could the authors explain the reason behind this? Can this be generalized and provide some guidance on how to choose the number of bits in the future applications?

Response:

Thank you for your comment.

We recognize the observation in our experiments where the watermark extraction accuracy exhibits an inverted U-shape. This phenomenon indicates that both too few and too many bits in the embedded watermark information can lead to vulnerability. When the number of bits is too low, the robustness of the watermark after encoding is diminished. As the number of bits increases, the accuracy does not significantly decrease. The slight decrease in accuracy might be attributed to the selection of experimental samples.

From our experimental findings, it seems that a watermark bit size of around 2000 is most suitable for the method discussed in our paper. However, it is important to consider the capacity of the watermark carrier in practical applications, as it determines the size of watermark information it can hold. We have included these explanations in the conclusion section of our paper to provide a comprehensive understanding of our experimental results and their implications.

Author action:

Page 13: In Chapter 3, we have included these explanations in the conclusion section of our paper to provide a comprehensive understanding of our experimental results and their implications.

  1. Comment: Could the authors provide more numerical results for the final paragraph in Section 3?

Response:

Thank you for your comment.

In the revised version of our paper, we have included a new experimental comparison between watermarking without the use of erasure codes for encoding watermark information and watermarking using the method proposed in our paper. We have also enhanced the section on experimental results and analysis, particularly the part discussing the conclusions, in the third section.

Author action:

Page 11: This study tested the robustness of semi-structured data watermarking against three types of attacks, applying varying degrees of typical attacks such as deletion, value erasure, and redundant insertion. The success rate of extracting the complete watermark from zero-watermarking was assessed, and a comparison was made with scenarios where watermark information was encoded without the use of erasure codes. For a fixed watermark size of 1000 bits, this paper conducted zero-watermark embedding and extraction tests using four different-sized JSON files: File 1, File 2, File 3, File 4, and File 5, each containing 100, 200, 800, 1500, and 3000 key-value pairs respectively after preprocessing and parsing. Multiple identical attacks were executed on each zero-watermark file. The success rates of watermark extraction under different attacks are presented in Table 1, where extractions with incorrectly retrieved bits are not considered successful. In the table, Pd, Pi, and Pm respectively indicate the percentage of key-value pairs that were deleted, inserted, or manipulated in the file. The results demonstrate that the method proposed in this paper significantly enhances the robustness of the watermark, especially against numerical tampering attacks.

Type of Attack

Direct extraction success rate / Extraction success rate using our method

Document 1

Document 2

Document 3

Document 4

Document 5

No Attack

1.0/1.0

1.0/1.0

1.0/1.0

1.0/1.0

1.0/1.0

Deleted Attack

Pd=5%

0.817/1.0

0.820/0.975

0.813/0.999

0.821/0.998

0.840/0.991

Pd=10%

0.610/0.957

0.554/0.995

0.653/0.997

0.541/0.950

0.687/0.981

Pd=20%

0.361/0.967

0.343/0.966

0.299/0.996

0.381/0.976

0.374/0.972

Pd=30%

0.121/0.958

0.201/0.984

0.176/0.976

0.141/0.992

0.187/0.981

Modified Attack

Pm=5%

0.671/0.986

0.487/1.0

0.578/0.984

0.398/0.990

0.633/0.989

Pm=10%

0.321/0.997

0.241/0.987

0.322/0.988

0.178/0.991

0.452/0.988

Pm=20%

0.167/0.998

0.101/0.984

0.221/0.979

0.089/0.986

0.281/0.995

Pm=30%

0.076/0.996

0.009/0.990

0.067/0.995

0.004/0.996

0.079/0.976

Inserted Attack

Pi=5%

0.876/0.999

0.859 /1.0

0.778/0.998

0.862/0.996

0.856/0.998

Pi=10%

0.675/0.988

0.685/0.987

0.574/0.997

0.681/0.979

0.591/0.997

Pi=20%

0.471/0.987

0.461/0.980

0.310/0.986

0.444/0.990

0.297/0.979

Pi=30%

0.223/0.988

0.291/0.987

0.177/0.979

0.187/0.984

0.141/0.990

Reviewer 2 Report

Comments and Suggestions for Authors

The paper presents a novel zero-watermarking method for tracking semi-structured power data. This method employs Robust Soliton Distribution (RSD) and Error Correction Codes (ECC) to generate intermediate sequences, aiming to enhance the robustness and security of zero-watermarking. Experimental results demonstrate that this method maintains data security while achieving a watermark extraction success rate of over 98%. The review comments of this paper are as follows:

1) The paper introduces a new approach in the field of zero-watermarking for semi-structured data, integrating Robust Soliton Distribution with Error Correction Codes. This is a notable innovation. However, the paper lacks a detailed explanation of the theoretical foundation behind RSD and ECC. It is recommended to include a more comprehensive discussion of these theories to help readers understand how they jointly contribute to the zero-watermarking technique.

2) The processes of data preprocessing, feature sequence extraction, and watermark information embedding mentioned in the paper are complex. It is suggested to provide a more intuitive flowchart or pseudocode to help readers clearly understand the entire process.

The experimental data and test scenarios used in the article need a more detailed description. For instance, specifics about the dataset, such as size, type, and source, and the configurations of the test scenarios, should be provided.

3) Although the paper mentions that the experiments demonstrate a watermark extraction success rate of over 98%, it lacks comparative analysis with other existing techniques. It is advised to include a comparison with at least one current technique to showcase the advantages of the proposed method.

4) More details about the experimental results are needed, including the performance under different types of attacks and performance evaluation with varying data scales.

Comments on the Quality of English Language

Fine.

Author Response

Original Manuscript ID: electronics-2839004

Original Article Title: “Robust Soliton Distribution-Based Zero-Watermarking for Semi-Structured Power Data”

Dear Editor-in-Chief:

Thank you for allowing a resubmission of our manuscript, with an opportunity to address the reviewers’ comments. Those comments are all valuable and very helpful for revising and improving our paper, as well as the important guiding significance to our research. Each suggested revision and comment, brought forward by the reviewers was accurately incorporated and considered. We have revised the content of the manuscript according to the valuable suggestions.

We highlighted all the revisions in red color. And the responses are highlighted in blue color.

Best regards,

Sincerely yours,

<Lei Zhao, Yunfeng Zou and Chao Xu> et al.

Response to reviewers

We gratefully thank the EIC and all reviewers for their time spent making their constructive remarks and useful suggestions, which has significantly raised the quality of the manuscript and has enabled us to improve the manuscript. Each suggested revision and comment, brought forward by the reviewers, was accurately incorporated and considered. Below the comments of the reviewers are response point by point and the revisions are indicated.

Reviewer #2:

  1. Comment: The paper introduces a new approach in the field of zero-watermarking for semi-structured data, integrating Robust Soliton Distribution with Error Correction Codes. This is a notable innovation. However, the paper lacks a detailed explanation of the theoretical foundation behind RSD and ECC. It is recommended to include a more comprehensive discussion of these theories to help readers understand how they jointly contribute to the zero-watermarking technique.

Response:

Thank you for your comment.

In response to your valuable feedback, we have revised our paper to include a more detailed explanation of the theoretical foundations behind Robust Soliton Distribution (RSD) and Error Correction Codes (ECC). We have added a discussion of these theories to enhance the reader's understanding of how they are integrated and contribute jointly to the zero-watermarking technique. This addition not only elucidates the underlying principles of our approach but also highlights the innovation and significance of integrating these two concepts in the field of zero-watermarking for semi-structured data. We believe these enhancements will greatly improve the clarity and depth of our paper. Thank you for your insightful suggestion.

Author action:

Page 9: In Chapter 2, Error Correction Codes (ECC) and Robust Soliton Distribution (RSD) are two critical concepts in the realm of digital communication and data encoding. ECC encompasses a set of techniques used to detect and rectify errors in transmitted data, ensuring its integrity and accuracy over noisy channels or unreliable storage. These codes, such as Reed-Solomon and Hamming codes, add redundant data to the original message, allowing for the correction of errors at the receiver’s end. While ECC focuses on error correction within a fixed-size data block, RSD optimizes the transmission process in dynamic communication environments, making them both indispensable for reliable and efficient digital communications.

  1. Comment: The processes of data preprocessing, feature sequence extraction, and watermark information embedding mentioned in the paper are complex. It is suggested to provide a more intuitive flowchart or pseudocode to help readers clearly understand the entire process.

Response:

Thank you for your comment.

In response to your suggestion, we have carefully considered the complexity of the processes of data preprocessing, feature sequence extraction, and watermark information embedding as mentioned in our paper. We have already provided a detailed flowchart that illustrates the watermark embedding and extraction processes. We believe that this flowchart effectively conveys the necessary information in an intuitive manner. While we acknowledge the potential benefits of adding pseudocode, we currently perceive it as somewhat redundant given the clarity provided by the existing flowchart. Therefore, we have not made any modifications in this regard. However, we are open to revisiting this decision and adding pseudocode in the future if it becomes apparent that it would further enhance the understanding of our processes. We appreciate your feedback and are committed to making our work as accessible and comprehensible as possible.

  1. Comment: The experimental data and test scenarios used in the article need a more detailed description. For instance, specifics about the dataset, such as size, type, and source, and the configurations of the test scenarios, should be provided.

Response:

Thank you for your comment.

In response to your feedback regarding the need for a more detailed description of the experimental data and test scenarios used in our article, we have taken action to address this concern. We have revised the section on experimental setup in our paper to include more specific and detailed information. This includes the size, type, and source of the dataset used, as well as the configurations of the test scenarios. We believe that these enhancements will provide a clearer and more comprehensive understanding of our experimental approach and context. We are committed to ensuring that our research is presented with the necessary detail to allow for thorough comprehension and reproducibility. We've highlighted these sections.

Thank you for highlighting this area for improvement.

Author action:

Thank you for your comment.

Page 11: We've highlighted these sections in Chapter 3 .

  1. Comment: Although the paper mentions that the experiments demonstrate a watermark extraction success rate of over 98%, it lacks comparative analysis with other existing techniques. It is advised to include a comparison with at least one current technique to showcase the advantages of the proposed method.

Response:

Thank you for your comment.

In the revised version of our paper, we have included a new experimental comparison between watermarking without the use of erasure codes for encoding watermark information and watermarking using the method proposed in our paper. We have also enhanced the section on experimental results and analysis, particularly the part discussing the conclusions, in the third section.

Author action:

Page 11: This study tested the robustness of semi-structured data watermarking against three types of attacks, applying varying degrees of typical attacks such as deletion, value erasure, and redundant insertion. The success rate of extracting the complete watermark from zero-watermarking was assessed, and a comparison was made with scenarios where watermark information was encoded without the use of erasure codes. For a fixed watermark size of 1000 bits, this paper conducted zero-watermark embedding and extraction tests using four different-sized JSON files: File 1, File 2, File 3, File 4, and File 5, each containing 100, 200, 800, 1500, and 3000 key-value pairs respectively after preprocessing and parsing. Multiple identical attacks were executed on each zero-watermark file. The success rates of watermark extraction under different attacks are presented in Table 1, where extractions with incorrectly retrieved bits are not considered successful. In the table, Pd, Pi, and Pm respectively indicate the percentage of key-value pairs that were deleted, inserted, or manipulated in the file. The results demonstrate that the method proposed in this paper significantly enhances the robustness of the watermark, especially against numerical tampering attacks.

Type of Attack

Direct extraction success rate / Extraction success rate using our method

Document 1

Document 2

Document 3

Document 4

Document 5

No Attack

1.0/1.0

1.0/1.0

1.0/1.0

1.0/1.0

1.0/1.0

Deleted Attack

Pd=5%

0.817/1.0

0.820/0.975

0.813/0.999

0.821/0.998

0.840/0.991

Pd=10%

0.610/0.957

0.554/0.995

0.653/0.997

0.541/0.950

0.687/0.981

Pd=20%

0.361/0.967

0.343/0.966

0.299/0.996

0.381/0.976

0.374/0.972

Pd=30%

0.121/0.958

0.201/0.984

0.176/0.976

0.141/0.992

0.187/0.981

Modified Attack

Pm=5%

0.671/0.986

0.487/1.0

0.578/0.984

0.398/0.990

0.633/0.989

Pm=10%

0.321/0.997

0.241/0.987

0.322/0.988

0.178/0.991

0.452/0.988

Pm=20%

0.167/0.998

0.101/0.984

0.221/0.979

0.089/0.986

0.281/0.995

Pm=30%

0.076/0.996

0.009/0.990

0.067/0.995

0.004/0.996

0.079/0.976

Inserted Attack

Pi=5%

0.876/0.999

0.859 /1.0

0.778/0.998

0.862/0.996

0.856/0.998

Pi=10%

0.675/0.988

0.685/0.987

0.574/0.997

0.681/0.979

0.591/0.997

Pi=20%

0.471/0.987

0.461/0.980

0.310/0.986

0.444/0.990

0.297/0.979

Pi=30%

0.223/0.988

0.291/0.987

0.177/0.979

0.187/0.984

0.141/0.990

  1. Comment: More details about the experimental results are needed, including the performance under different types of attacks and performance evaluation with varying data scales.

Response:

Thank you for your comment.

In light of your feedback requesting more detailed experimental results, including performance under different types of attacks and performance evaluation with varying data scales, we have now included these details in our paper. Our expanded results section now provides a thorough performance analysis for different attack scenarios and evaluates how our proposed method scales with various data sizes. Thank you for guiding us to improve the completeness of our research presentation.

Author action:

Page 13 : In Chapter 3, we have provided a revised description, and We added a new set of comparison experiments to enhance our results.

Table 2. Rate of successful extraction under different combination attacks

type of attack

Direct extraction success rate / Extraction success rate using our method

Document 1

Document 2

Document 3

(Pd=10%,Pm=10%)

0.234/0.999

0.351/1.0

0.283/0.998

(Pd=10%,Pi=10%)

0.132/0.988

0.187/0.987

0.236/0.997

(Pm=10%,Pi=10%)

0.228/0.987

0.190/0.980

0.310/0.986
